# Atom Identifiers Generated by a Neighborhood-Specific Graph Coloring Method Enable Compound Harmonization across Metabolic Databases

**DOI:** 10.3390/metabo10090368

**Published:** 2020-09-11

**Authors:** Huan Jin, Joshua M. Mitchell, Hunter N. B. Moseley

**Affiliations:** 1Department of Toxicology and Cancer Biology, University of Kentucky, Lexington, KY 40536, USA; huan.jin@uky.edu; 2Department of Molecular & Cellular Biochemistry, University of Kentucky, Lexington, KY 40536, USA; jmmi243@uky.edu; 3Markey Cancer Center, University of Kentucky, Lexington, KY 40536, USA; 4Resource Center for Stable Isotope Resolved Metabolomics, University of Kentucky, Lexington, KY 40536, USA; 5Institute for Biomedical Informatics, University of Kentucky, Lexington, KY 40536, USA

**Keywords:** metabolomics, atom-resolved metabolic network, atom identifier, compound identifier, database harmonization, graph theory, common subgraph isomorphism

## Abstract

Metabolic flux analysis requires both a reliable metabolic model and reliable metabolic profiles in characterizing metabolic reprogramming. Advances in analytic methodologies enable production of high-quality metabolomics datasets capturing isotopic flux. However, useful metabolic models can be difficult to derive due to the lack of relatively complete atom-resolved metabolic networks for a variety of organisms, including human. Here, we developed a neighborhood-specific graph coloring method that creates unique identifiers for each atom in a compound facilitating construction of an atom-resolved metabolic network. What is more, this method is guaranteed to generate the same identifier for symmetric atoms, enabling automatic identification of possible additional mappings caused by molecular symmetry. Furthermore, a compound coloring identifier derived from the corresponding atom coloring identifiers can be used for compound harmonization across various metabolic network databases, which is an essential first step in network integration. With the compound coloring identifiers, 8865 correspondences between KEGG (Kyoto Encyclopedia of Genes and Genomes) and MetaCyc compounds are detected, with 5451 of them confirmed by other identifiers provided by the two databases. In addition, we found that the Enzyme Commission numbers (EC) of reactions can be used to validate possible correspondence pairs, with 1848 unconfirmed pairs validated by commonality in reaction ECs. Moreover, we were able to detect various issues and errors with compound representation in KEGG and MetaCyc databases by compound coloring identifiers, demonstrating the usefulness of this methodology for database curation.

## 1. Introduction

Metabolic flux analysis is an essential approach to access metabolic phenotypes [1,2] that requires both reliable metabolic profiles as well as reliable metabolic models [3,4,5]. Advances in analytical technologies like mass spectrometry (MS) and nuclear magnetic resonance (NMR) greatly contribute to the detection of thousands of metabolites from biofluids, cells, and tissues [6]. Application of those analytical techniques to stable isotope resolved metabolomics (SIRM) experiments facilitates production of high-quality metabolomics datasets capturing isotopic flux through cellular and systemic metabolism [7,8]. Now, the challenge is to construct meaningful metabolic models from the corresponding metabolic profiles for downstream metabolic flux analysis. A metabolic network is usually represented by compounds connected via biotransformation routes [9]. Obviously, information at the atom level is not represented in such metabolic networks, making it impractical to derive appropriate metabolic models for SIRM datasets. Prior work demonstrated an atom-resolved metabolic network that included both central and intermediate metabolism in *Escherichia coli* that allowed atom-to-atom tracing [10,11]. However, currently there are no relatively complete atom-resolved databases of metabolic networks available for human metabolism that can be used to trace individual atoms [12]. 

To construct an atom-resolved metabolic network, compounds and metabolic reactions with detailed documentation at the atom level are required. One approach is to reconstruct a hypothetical atom-resolved metabolic network from generalized reaction descriptions that are atom-specific [13]. However, it is unclear what level of validation and curation such an approach would require to construct a reasonably accurate atom-resolved metabolic network for generating metabolic models usable in the analysis of SIRM datasets. An alternative is to use curated metabolic databases currently available, in particular the Kyoto Encyclopedia of Genes and Genomes (KEGG) and the MetaCyc metabolic pathway database. The popular molfile description of a compound is a text-based chemical table file format developed by MDL Information Systems and contains information about atoms, bonds, connectivity, and coordinates [14], which is available in most databases including KEGG and MetaCyc. For atom-resolved metabolic reactions, the KEGG reaction pair (RPAIR) database stores patterns of transformations occurring between two reactants in a single reaction [15]. In addition, MetaCyc contains direct atom mappings for every metabolic reaction [16]. Previous work only made use of atom mappings in either the KEGG RPAIR database [17,18] or MetaCyc [19] for atom tracing. However, both databases cover metabolism for many common organisms, clearly indicating that these two databases are not independent of each other. A necessary first step for constructing a more comprehensive network is to integrate compounds from different databases without redundancy [20].

In an atom-resolved metabolic network, each node should include information at both molecule-specific and atom-specific levels. To name each atom in a compound, two rules need to be obeyed: (1) different atoms must have different identifiers; (2) symmetric atoms must share the same identifier. Previous work used the atom index in the molfile associated with a compound in finding atom-specific metabolic pathways without considering molecular symmetry [18,19]. Likewise, molecular symmetry has been ignored in prior atom-resolved metabolic network reconstruction approaches [13]. One group tried to assign a unique name for every atom in the compound based on the compound’s International Union of Pure and Applied Chemistry (IUPAC) International Chemical Identifier (InChI) representation [21], which does not apply to this scenario since symmetric atoms can share the same routes in the metabolic network. In addition, any InChI-based approach cannot handle the compound entries with R-groups. To our knowledge, no appropriate method has been previously published that provides each atom in a compound with a useful identifier for the explicit purpose of constructing an atom-resolved metabolic network, either because the identifier was not unique or because it was not consistent for symmetric atoms.

In this paper, we developed a novel neighborhood-specific graph coloring method that creates a unique identifier for each atom in a compound by expanding the type (color) of each atom based on its “neighborhood” of atoms (nodes) bonded (edges) to it. This approach is related to but distinct from atom typing performed in chemoinformatics, which determines an augmented atom type based on the local chemical environment, especially the directed bonded atoms [22]. Atom coloring creates an augmented atom type based on both directly and indirectly bonded atoms that are part of the graph neighborhood around a given atom. Moreover, the method is guaranteed to generate the same coloring identifier for symmetric atoms. Furthermore, compound coloring identifiers derived from the corresponding atom coloring identifiers can be used for compound harmonization across metabolic databases. In this context, only molecular configuration (i.e., changes requiring the breaking of a bond) and not molecular conformation (i.e., changes not requiring the breaking of a bond like a bond rotation) are considered in the generation of these identifiers. To our knowledge, this is the first attempt to create unique atom and compound identifiers that are consistent with respect to molecular symmetry and for the explicit purpose of harmonizing compounds across the KEGG and MetaCyc databases, ultimately to facilitate the construction of an integrated atom-resolved metabolic network. 

## 2. Results

### 2.1. Overview of KEGG and MetaCyc Databases

The numbers of compounds and atom-resolved reactions in KEGG and MetaCyc databases are summarized in Table 1. MetaCyc has 1.09 times as many compound entries as KEGG and 1.53 times as many atom-resolved reaction entries.

To initially evaluate the level of overlap between KEGG and MetaCyc databases, we used existing identifiers in each database to find the correspondences between KEGG and MetaCyc compounds. Not all compounds in either database have all the chemical identifiers listed in Table 2. Some compounds in MetaCyc have a direct identifier to the corresponding KEGG compound [19]. We can see that the number of matched compounds (correspondences) detected by different identifiers are not consistent, with a total of less than 5700. We also generated InChI identifiers based on the molfile provided for each entry in each database using Open Babel [23], which utilizes the InChI software library provided by the InChI Trust [24]. We were able to generate 16,530 InChI from KEGG and 15,765 InChI from MetaCyc, providing 3103 correspondences. When combined with ChEBI and KEGG Compound IDs, a total of 5929 consistent correspondences were detected. Two issues may appear when applying these identifiers to compound integration across various databases. On the one hand, there is no easy way to check if some correspondences are missing. In addition, it is difficult to tell if the results generated by those identifiers are correct, since errors can exist in every database [21,25]. Such errors are illustrated by the 964 out of 13,216 KEGG compound entries with InChI that are inconsistent with the InChI generated from their associated molfile, representing 7.3% of the InChI-containing entries in KEGG. Likewise, 55 out of 15,076 MetaCyc compound entries have InChI that are inconsistent with the InChI generated from their associated molfile, representing 0.4% of the InChI-containing entries in MetaCyc. 

Therefore, a reliable, systematic naming method for chemical compounds that solves problems at the atom-level as well as at the compound-level is required for constructing an atom-resolved metabolic network. Towards this end, we have developed a neighborhood-specific graph coloring method (see Section 4.6) that derives unique identifiers for atoms as well as compounds.

### 2.2. Aromatic Substructure Detection

The neighborhood-specific graph coloring method described in Section 4.6 is very sensitive to the specific structural representation. Moreover, aromatic substructures are not consistently represented in both databases. Instead of being directly labeled as an aromatic bond type, single and double bonds are used alternatively to depict the aromatic substructure. CPD-6962 in MetaCyc has a direct reference KEGG compound C15523 (Figure 1). We can see that the positions of double bonds and single bonds within the benzene ring vary between these two representations, which can lead to two different sets of atom identifiers. Therefore, we needed to ensure that compound representation is consistent across databases so that each compound will have a single set of atom identifiers. In this case, we first detect the aromatic substructures in all compounds from both databases, and change the single and double bonds within the aromatic substructure to aromatic bond.

Two independent aromatic detection methods were used in aromatic substructure detection: our Biochemically Aware Substructure Search (BASS) method [26] which uses neighborhood-specific graph coloring [27] to greatly improve subgraph isomorphism detection [28] and the aromatic detection facilities in the Indigo package [29]. First, we compared the aromatic substructures derived by these two methods. As shown in Table 3, Indigo appears more conservative than BASS in detecting aromatic substructures, detecting roughly 85% of what the BASS method does. Figure 2 shows an example of an aromatic substructure that can be missed by Indigo. We assume that Indigo cannot detect aromatic substructures with a double bond connected to atoms outside of the ring. This is not surprising, since BASS leverages the curated set of aromatic substructures in KEGG and has very high precision (99.9%) in the detection of aromatic substructures in KEGG compounds, while Indigo uses a set of simplified aromatic detection heuristics along with hard-coded algorithmic limitations of ring sizes being searched. However, we had concerns that some valid aromatic substructure representations in MetaCyc compounds may not exist in the reference aromatic substructure set derived from the KEGG database, which would be missed by the BASS method. This was confirmed by Indigo detecting 30 additional MetaCyc compounds with aromatic substructures not detected by the BASS method. Therefore, we combined the KEGG aromatic substructures with additional Indigo-detected substructures from MetaCyc. By using both methods, we were able to detect aromatic substructures in about half of the compounds in each database (Table 4). When an aromatic substructure was detected, all bonds for the aromatic substructure were changed to an aromatic bond type and the modified molfile was saved. All analyses were performed on a desktop computer with an i7-6850K CPU (6-core with HT), 64 GB RAM, and 512 GB solid state drive. On this hardware, the aromatic substructure detection took less than 5 min for KEGG and roughly 15 min for MetaCyc in terms of execution time.

### 2.3. Generating Identifiers for Atoms Using a Graph Coloring Method

Since symmetric atoms share the same neighbors, the graph coloring method is guaranteed to create the same identifier for them. Our concern is whether atoms with the same identifier are actually symmetric. In our graph coloring method, we only include 0_layer identifiers (see Section 4.6 for graph coloring method) in atom coloring to avoid long name strings. In some extreme cases, this shortcut can assign the same identifier to atoms that are asymmetric. An example is shown in Figure 3A. We can see that this compound does not contain any symmetric atoms. Without considering the upper right ring, the bottom two rings are symmetric. Therefore, atom 1 and 2 have the same 0_layer identifier, which is the same for atom pairs 4 and 5 and 6 and 7. In addition, once atom 1 and 2 reach atom 3, they will share the same route to the upper right substructure. Finally, atom 1 and 2 will share the same coloring identifier (Figure 3B) even though they are not symmetric. To deal with this problem, atom coloring validation and recoloring is performed. We can see that atom 1 and 2 have distinct identifiers after recoloring (Figure 3C).

We validate symmetry after a first round of coloring, recolor the compound if asymmetric atoms have the same identifier, and verify symmetry again. After this coloring-validation-recoloring-validation process, our results indicated that the graph coloring method is able to generate the same identifier for symmetric atoms and asymmetric atoms have unique identifiers for all compounds in both KEGG and MetaCyc databases.

### 2.4. Detection of Correspondences Between KEGG and MetaCyc Compounds via Coloring Identifiers

After creating a single set of atom identifiers for each compound, we were able to derive ordered compound coloring identifiers at different levels of chemical specificity, which can be used to harmonize compounds across databases. Since KEGG and MetaCyc can include different numbers of H (hydrogen atoms) in the molfile, we exclude H in coloring at this point. We first tried to include information of bond stereochemistry, atom charge, atom stereochemistry, and isotope stereochemistry in coloring to ensure each compound has a unique name. With the relatively specific coloring identifiers, 1762 correspondences between KEGG and MetaCyc compounds can be detected (see Table 5), which is not satisfactory compared to 5681 pairs discovered by other identifiers (e.g., KEGG, CHEBI, and InChI as shown in Table 2). This lack of correspondence is due to the inconsistencies in bond stereochemistry, atom charge, atom stereochemistry, and isotope stereochemistry information between these two databases. An example is shown in Figure 4, where compound CPD-20570 in MetaCyc has a direct reference to KEGG compound C13014. 

For the following analysis, we only included information of atom type and bond type to keep the backbone of a compound in atom naming. It took less than 10 min of execution time on a desktop computer built by System76 (Denver, CO, USA) with an Intel i7-6850K CPU (6-core with HT), 64 GB RAM, and a 512 GB solid state drive to generate these coloring identifiers for all compound entries in the KEGG and MetaCyc databases. About 8865 correspondences between KEGG and MetaCyc are detected (see Table 5 and Appendix A), and 5451 of them can be confirmed by other identifiers. With both tight and loose compound coloring identifiers, about 95.95% compounds pairs detected by other chemical IDs can be discovered. We manually checked the compound pairs that were discordant with other chemical IDs and found that none of them are caused by an inconsistency between the coloring identifier and the compound representation. Appendix A uses the MetaCyc CPD-20570 and KEGG C13014 as an example, illustrating how loose coloring addresses the issues caused by tight coloring (Figure 4) and facilitates compound harmonization. The question then becomes how to validate the remaining 3414 possible pairs. Matched compounds are supposed to take part in the same metabolic reactions. The Enzyme Commission (EC) number is a numerical classification scheme for enzymes, playing a key role in classifying enzymatic reactions [30,31]. We expected matched compounds to take part in metabolic reactions with similar EC numbers. 

Then, we analyzed the metabolic reactions in KEGG and MetaCyc databases (see Table 6). We can see that the documentation of EC number in KEGG is more complete compared to MetaCyc, but the number of metabolic reactions in MetaCyc is 50% larger than in KEGG. Around 80% of reactions in both databases can be related to at least a 3-leveled EC number.

Next, we tested how well EC numbers work in the validation of correspondences between KEGG and MetaCyc compounds (see Table 7). We first identified color-harmonized pairs that both take part in some reactions in their respective database. There are 4227 ID confirmed pairs and 2292 possible pairs involved in the metabolic reactions. We further investigated if those pairs participate into the same type of reaction indicated by EC number. If we use the first 3 levels of the sectioned EC number as the standard, 3810 (90.13%) ID-confirmed pairs are verified by 3-leveled EC numbers. In addition, 3580 of them can be further confirmed by 4-leveled EC numbers. Furthermore, 1848 and 1540 possible pairs are confirmed by 3-leveled and 4-leveled EC numbers, respectively. These results suggest that EC numbers may be useful in validating possible pairs that have slight coloring deviations. All of the detected compound pairs are listed in Appendix A.

### 2.5. Compound Representation Errors and Issues Detected in the KEGG and MetaCyc Databases

When harmonizing compounds between KEGG and MetaCyc databases, we found that there are various compound representation issues and errors existing in both databases, which can be grouped into several categories like mismatch between compound image and molfile, incorrect cross-referencing, and different bonds attached to metal ions. Here, we give a brief description with some examples, and all the detected inconsistency is documented in Appendix A.

#### 2.5.1. Incomplete KEGG Aromatic Atom Types

KEGG atom types annotate every atom in every compound of the KEGG Compound database. The KEGG atom type of an atom maps that atom to a unique chemical substructure and these substructures often map to functional groups (e.g., the atom type “O1a” represents an oxygen of a hydroxyl group). However, the set of KEGG atom types is not complete, especially with regard to aromatic heterocycle atoms. In particular, there are no oxygen and sulfur aromatic KEGG atom types defined, which prevents full automation of aromatic substructure determination based on KEGG atom type alone. We used a simple heuristic method (i.e., a simple deterministic decisioning approach) to consider oxygen and sulfur atoms as aromatic when they are part of a ring where all other carbon and nitrogen atoms are labeled as aromatic, based on KEGG atom types. However, this aromatic substructure detection approach has limitations that require some manual inspection, as highlighted in Figure 5. KEGG Compound entry C03861 contains a 1,4-dioxin flanked by aromatic rings. The 1,4-dioxin is not aromatic. In a counter-example KEGG Compound entry C07729 contains an aromatic pyridine substructure flanked by benzyl rings. The presence of both examples illustrates why aromatic substructure detection cannot be fully automated based on the current set of KEGG aromatic atom types. In addition, Appendix A shows a KEGG compound with an S-containing aromatic ring.

As an aside, the quinoid fragment in KEGG Compound entry C03861 is likely mislabeled as aromatic, since quinoid fragments are standardly antiaromatic [32]. This quinoid fragment was likely mislabeled as aromatic due to the whole three-ring structure obeying Huckel’s rule. While we treated KEGG-identified aromatic substructures as completely correct, this example does indicate the presence of some error in KEGG’s aromatic substructure detection methods. Comparison of Indigo to KEGG may provide a means for detecting suspect KEGG aromatic substructures, but a manual inspection of all suspect substructures is not practical, especially from an automated analysis perspective. Moreover, aromatic mislabeling should not impact compound harmonization if applied consistently across databases. 

#### 2.5.2. Inconsistent Compound Representations

Using ID-based compound harmonization, we found that there are about 10 MetaCyc compounds that contain valid aromatic substructures not detected by either the BASS or Indigo methods (Figure 6). To deal with this problem, we incorporated those valid aromatic substructures into the reference aromatic substructure set.

#### 2.5.3. Incorrect Cross-Referencing

There are some matched compounds detected by other identifiers that do not have the same coloring identifier. Compound CPD-19437 in MetaCyc has a direct reference to KEGG compound C12187, but their coloring identifiers are different (see Figure 7). We can see that the compound representation in MetaCyc is not consistent with its counterpart in KEGG. In addition, CPD-19437 and C12187 have the same ChEBI reference compound 32074, and the representation in ChEBI is the same with that of KEGG, suggesting the representation in MetaCyc may be incorrect.

### 2.6. Estimating the Error Rate of the Graph Coloring Method

#### 2.6.1. Ambiguous Coloring Identifiers

During the compound harmonization process, tight atom and compound coloring was loosened (see Figure 4 for an example), keeping only atom type and bond type in the atom coloring for the final steps in compound harmonization. With the loose coloring, multiple compounds in one database can have the same coloring identifier. We first tested if a compound can have a unique coloring identifier when all information is included in the atom coloring with hydrogen (H) atoms excluded (Table 8). Here, we did not count compounds with a generic R group representing ambiguous functional groups and substructures; however, the results that include all compounds are described in Appendix A. Several types of compounds cannot be distinguished by the tight coloring identifier except for those duplicates (see Appendix A). When we only include atom type and bond type in the atom coloring, many more compounds share the same coloring identifier. After compound harmonization, we are able to detect compounds with the same coloring identifier from the source database.

When the compound identifier is ambiguous, a compound in one database can be mapped to several different compounds in the other database during compound harmonization. For ID confirmed pairs, 28 MetaCyc compounds can be linked to more than one KEGG compound, which is caused by inconsistency of different ID references. In addition, about 478 MetaCyc compounds have several KEGG correspondences among the 1848 pairs verified by 3-leveled EC. This highlights the value in leveraging metabolic reactions and the corresponding atom mappings to disambiguate multiple possible mappings while constructing an integrated metabolic network.

#### 2.6.2. Pseudosymmetric Atoms

Omitting information in the atom coloring can also lead to pseudosymmetric atoms. We tested if incorporation of atom charge, atom stereochemistry, or bond stereochemistry in the atom coloring will erase some symmetric atoms (Table 9). After addition of atom charge, 148 MetaCyc and 38 KEGG compounds lose symmetry. Most of them are caused by terminal atoms, like CPD-321 (Figure 8). Since either symmetric atom can be labeled with charge, asymmetry caused by atom charge can be ignored in constructing a metabolic network. In addition, both databases contain compounds affected by bond and atom stereochemistry. We need to take bond and atom stereochemistry into consideration, since some enzymes are stereochemically specific. A heuristic method could be used to test if symmetric atoms are affected by bond and atom stereochemistry, and then atom coloring identifiers incorporated with bond and atom stereochemistry will be generated to overcome this issue. However, more complex molecular symmetries like those illustrated by KEGG C04167 will require the use of algorithms that can detect all possible molecular symmetries (i.e., automorphisms induced by rotations and reflections of the ℜ^3^ embedded graph) using a 3-dimensional representation of the compound [33].

#### 2.6.3. Changeable Graph Representation

There are two types of matched compounds that cannot be detected by coloring identifiers. One group of compounds can have either linear or circular representations (see Figure 9), and there are about 26 examples in this category. The other group is caused by resonance structures (see Figure 10), and we discovered about 46 similar cases. Artificial sets of atom mappings can be created to represent chemical transformations that are spontaneous.

## 3. Discussion

Here, we have developed a graph coloring method that creates unique identifiers for each atom in a compound with consideration for molecular symmetry. The atom-specific identifiers can capture additional cross-reaction atom mappings caused by symmetric atoms, which will contribute to the construction of a more complete atom-resolved metabolic network requiring information at both the compound and atom levels. Towards this overall goal, the ordered compound coloring identifiers derived from the corresponding atom coloring identifiers facilitate compound harmonization across metabolic databases, which is an essential first step in cross-database network integration. Different databases can have a distinct preference in compound representations, especially for aromatic substructures. To overcome inconsistent aromatic representations between databases, we devised a pragmatic BASS method [26] for aromatic substructure detection that leverages the labeled aromatic substructures in KEGG. Application of BASS to KEGG validated the method, providing confidence in its application to the MetaCyc database. The automatic aromatic atom detection method in Indigo [29] further validated the comprehensiveness of our BASS aromatic substructure detection method, which leverages KEGG’s curated aromatic substructures. The combination of BASS and Indigo can achieve good performance in aromatic substructure detection. This was further augmented by detecting additional aromatic substructure representations in MetaCyc through ID-based compound harmonization. In addition, compound states such as atom charge are not always the same between KEGG and MetaCyc. Therefore, identifiers like InChI that include these details to achieve an unambiguous label are not a good choice for maximizing cross-database compound harmonization in this situation. Furthermore, InChI cannot handle the compound entries that contain R-groups. However, InChI is very useful for validation of the presented methods’ development. Simplified molecular-input line-entry system (SMILES) identifiers and its derivatives are not a good option, because SMILES and its derivatives are not guaranteed to generate a unique identifier. In addition, neither InChI nor SMILES deal with the unique naming of atoms that is consistent for symmetric atoms. While the molecular graph coloring method has similarities to molecular canonicalization methods [24,34,35], it was designed to facilitate harmonization of compounds between metabolic databases. The graph coloring method is flexible in adjusting information used in atom coloring, which can help detect more possible matched compounds with a higher false positive rate. With the coloring identifiers, we were able to detect 8865 correspondences between KEGG and MetaCyc compounds, and 5451 of them can be confirmed by other identifiers. In addition, commonality in EC numbers associated with reactions and compounds provided another avenue for both validating and predicting possible correspondence pairs. This method validated 1848 pairs unconfirmed by other identifiers. While harmonizing compounds between KEGG and MetaCyc, we detected various issues and errors in the databases by coloring identifiers which are enumerated in the Appendix A, suggesting that this method can also be used for curation of current metabolic databases. Furthermore, the graph coloring method and compound harmonization approach can be used to integrate any metabolic database that provides a molfile representation of compounds, greatly facilitating future construction of more complete integrated metabolic networks.

## 4. Materials and Methods 

### 4.1. Compound and Metabolic Reaction Data

All data were downloaded directly from the corresponding databases. The KEGG COMPOUND and KEGG REACTION data is from the version available from KEGG on May 2019 via its REST interface. MetaCyc compound and reaction data is in version 23.0, downloaded from BioCyc.

### 4.2. Overview of Major Analysis Steps

A compound can be represented as a graph where each node is an atom in the compound and each edge between atoms is a chemical bond. Based on the molfile, we are able to create a graph representation for the corresponding compound. After we detect the aromatic substructures for a compound, we can change the bonds within the aromatic substructures to aromatic type (molfile [14] bond designation 4). After curation of aromatic substructures and double bond stereochemistry, we performed atom coloring and validation to guarantee that symmetric atoms share the same identifier and different atoms have different identifiers. Each set of atom identifiers for a compound is used to derive the corresponding compound coloring identifier. Finally, we detect corresponded pairs of compounds across two databases using ordered compound identifiers for each compound in each database. The flowchart of the overall compound harmonization procedure is shown in Figure 11.

### 4.3. Molfile Parser

We used a modified ctfile Python 3 package [36] to parse a molfile into atom and bond blocks, and saved them into the JavaScript Object Notation (JSON) format [37], facilitating access and modification.

### 4.4. Aromatic Substructure Detection

We used two methods in aromatic substructure detection. One is based on common subgraph isomorphism detection, and the other is an automatic aromatic atom detection method in Indigo packages [29]. In the KEGG database, aromatic atoms in a compound are specified in its KEGG Chemical Function (KCF) file [38]. Based on the aromatic atoms, we were able to extract the aromatic substructures present within a compound, and then saved every substructure into a separate molfile. If several aromatic rings are connected, we would fuse them together as one substructure. Then, we built a set of all aromatic substructures detected from the KEGG compounds without duplication. Furthermore, we manually inspected the set of aromatic substructures to ensure data quality. With this curated set of reference aromatic substructures, we tested each compound in a database for the presence of any of these aromatic substructures using the BASS method [26]. We analyzed KEGG to validate the aromatic substructure detection method itself. Then, we analyzed MetaCyc and labeled the bonds of detected aromatic substructures as aromatic. Furthermore, valid aromatic substructures in MetaCyc compounds can be detected by Indigo and other IDs. Finally, we created 366 KEGG-derived and 21 MetaCyc-derived aromatic substructures in the reference aromatic substructure set.

### 4.5. Identification of Double Bond Stereochemistry

The C = C double bond stereochemistry is not clearly specified in the molfile in both databases. To distinguish cis/trans stereoisomers, we adopted a method for automated identification of double bond stereochemistry [39]. This method requires fully hydrogenated compounds. Therefore, we first used Open Babel [23] to add hydrogen atoms for every compound, and then performed the calculation.

### 4.6. Neighborhood-Specific Graph Coloring Method

Our neighborhood-specific graph coloring method is based on a breadth-first search algorithm [40]. This method names each atom based on its own and neighbors’ chemical information, which can include atom type, atom charge, atom stereochemistry, isotope, bond type, and bond stereochemistry. The method is flexible in adjusting the chemical information included in the atom coloring. A flowchart of the graph coloring method is shown in Figure 12. First, the method names each atom with its own chemical information, which is saved as the 0_layer identifier and the start of the current atom identifier. Then, the method builds a dictionary that relates each atom with its 0_layer identifier and directly linked atoms. Directly bonded atoms of each atom are initialized as its neighbors. The method continues to extend the name of each atom, adding information about its neighbors into the 0_layer dictionary to its current identifier, and updating neighbors with neighbors’ neighbors that have not been used in extending the name of that atom. The method first repeats this process 3 times for all the atoms to avoid early stopping that can lead to non-unique compound coloring identifiers. Then, the method checks if an atom has a unique identifier. Atom naming will continue for those atoms that still share the same identifiers with other atoms until all the atoms in the compound have been used in name extension. Finally, the current name for each atom will be its coloring identifier. Compound C00047 in KEGG database (Appendix A) is used as an example to illustrate how the method works (Appendix A).

### 4.7. Atom Coloring Validation and Recolor

The atom coloring validation and recoloring are also based on a breadth-first search algorithm. The atom coloring validation flowchart is shown in Figure 13. For atoms with the same coloring identifier, we checked if neighbors of these atoms are also the same, layer by layer, until all the atoms in the compound have been tested. Then, the recoloring method corrects atoms with the same identifier that do not have the same neighbors. The recoloring process is similar to the graph coloring method. Instead of creating a 0_layer identifier dictionary, we use a full identifier dictionary. In addition, we only color atoms to where they have different neighbors to distinguish between them.

### 4.8. Creation of Compound Coloring Identifiers Based on Atom Coloring Identifiers

Once we create the identifiers for all the atoms in a compound, we can combine the number of atoms with the same identifier along with the atom coloring identifier. We sorted all the substrings, and then concatenated them together to form an ordered coloring identifier for the compound. The formulation is shown in Equation (1), which represents the order of string concatenation with *n_k_* being the number of atoms with coloring *a_k_*. The parenthesis and bracket characters are included in the resulting string.
(1)$$Compound\;color\;identifier\;=\;\left(n_{1}\right)\left[a_{1}\right]\left(n_{2}\right)\left[a_{2}\right]\left(n_{3}\right)\left[a_{3}\right]\;\ldots .\;\left(n_{k}\right)\left[a_{k}\right]$$

### 4.9. Prediction of Possible Compound Correspondence via Metabolic Reactions

We connected each compound with the metabolic reactions of which it is a part. For matched compounds between KEGG and MetaCyc, we tested if the compound shared at least one metabolic reaction indicated by the EC number in both databases.

## Figures and Tables

**Figure 1 metabolites-10-00368-f001:**
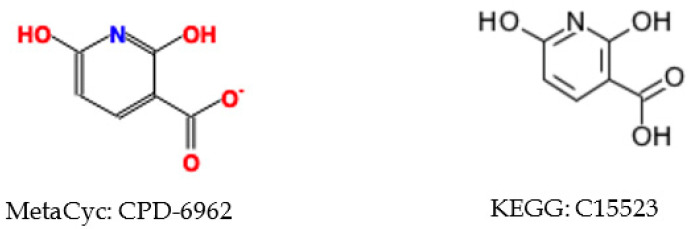
Correspondence between KEGG and MetaCyc compound entries with different molecular representations.

**Figure 2 metabolites-10-00368-f002:**
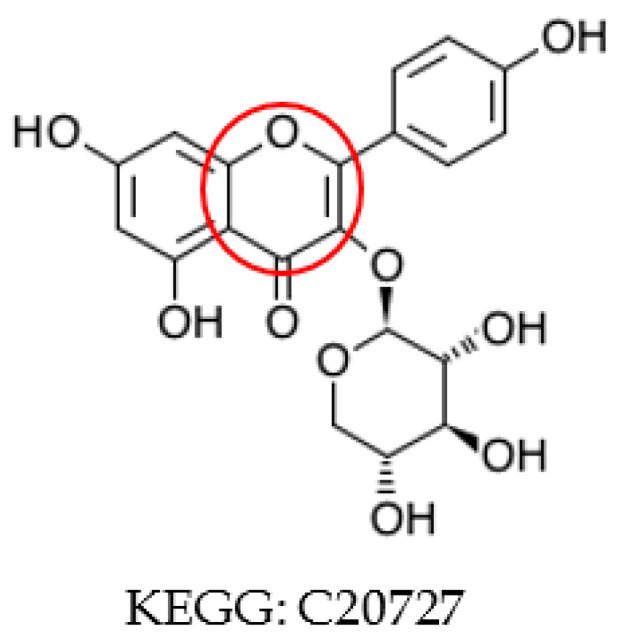
Example aromatic substructure that cannot be detected by Indigo.

**Figure 3 metabolites-10-00368-f003:**
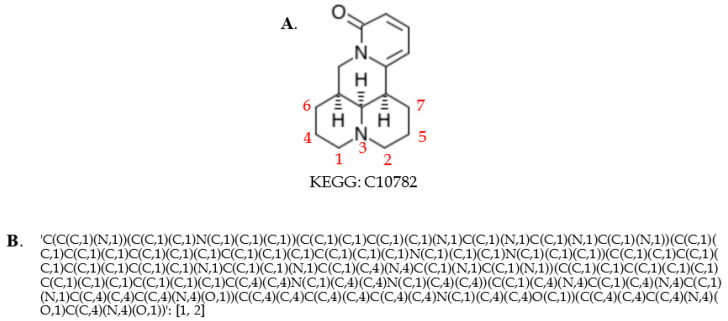
(**A**) Example of compound with same atom identifier for asymmetric atoms using an overly simplistic coloring approach; (**B**) the atom identifiers for atom 1 and 2 before symmetry validation; (**C**) the atom identifiers for atom 1 and 2 after symmetry curation.

**Figure 4 metabolites-10-00368-f004:**
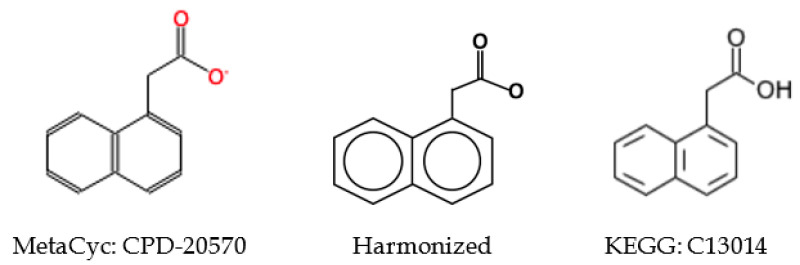
Example of charge inconsistency of compound representations between databases. The middle harmonized compound representation enables loose coloring that facilitates compound harmonization.

**Figure 5 metabolites-10-00368-f005:**
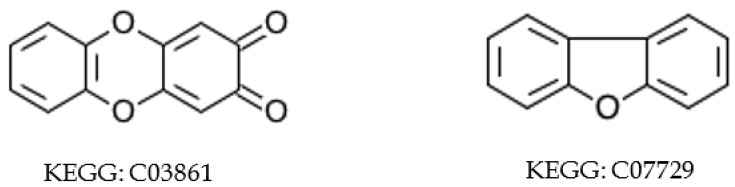
Compound with incomplete KEGG aromatic atom types. The middle ring of compound C03861 (**left**) is not aromatic while the middle ring of compound C07729 (**right**) is aromatic.

**Figure 6 metabolites-10-00368-f006:**
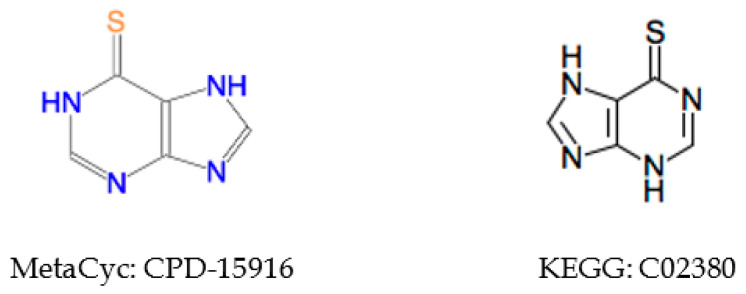
Compound with different aromatic representations. These two corresponding compound entries across KEGG and MetaCyc have two different aromatic substructure representations.

**Figure 7 metabolites-10-00368-f007:**
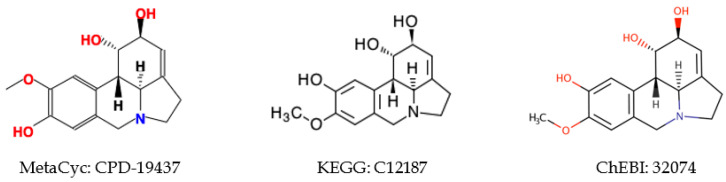
Example of inconsistent compound representations between KEGG and MetaCyc.

**Figure 8 metabolites-10-00368-f008:**
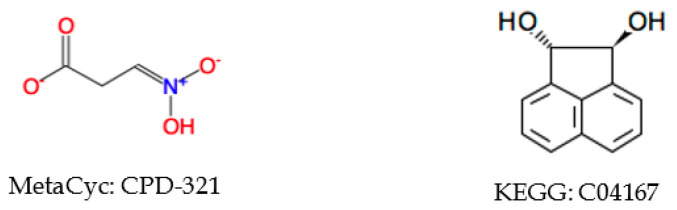
Examples of compounds gaining asymmetry after the addition of tight atom coloring information. For CPD-321, the two oxygens bound to the nitrogen are asymmetric with tight atom coloring and symmetric with loose atom coloring.

**Figure 9 metabolites-10-00368-f009:**
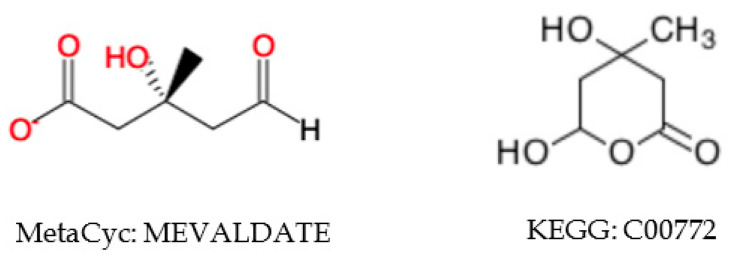
Compound with linear and circular representations.

**Figure 10 metabolites-10-00368-f010:**
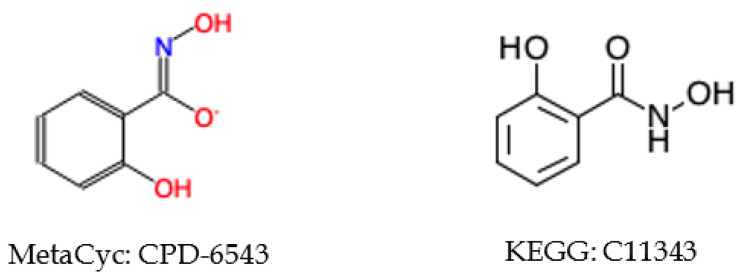
Compound with different resonance structures.

**Figure 11 metabolites-10-00368-f011:**
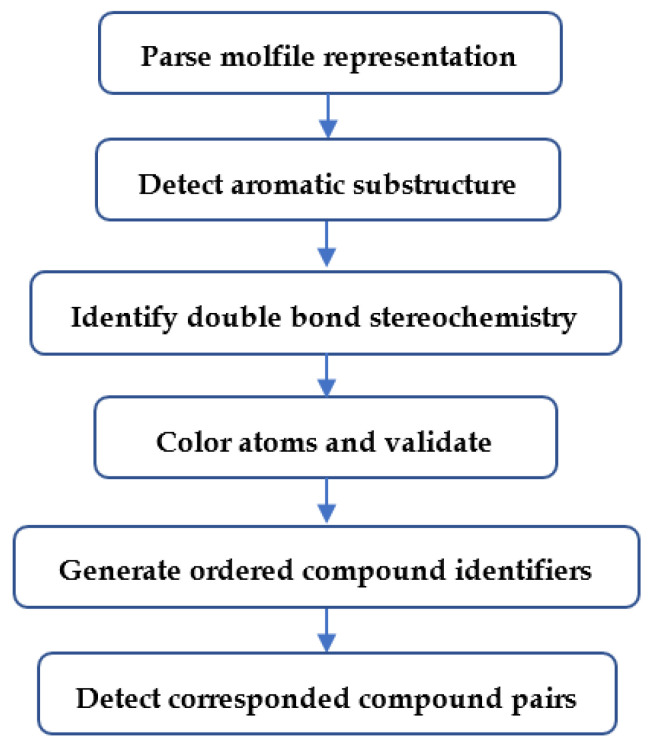
Overview of major compound harmonization steps.

**Figure 12 metabolites-10-00368-f012:**
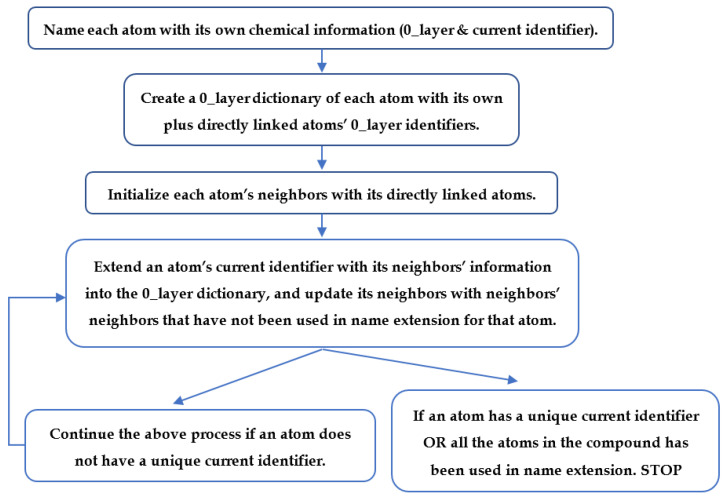
Flow chart of atom coloring.

**Figure 13 metabolites-10-00368-f013:**
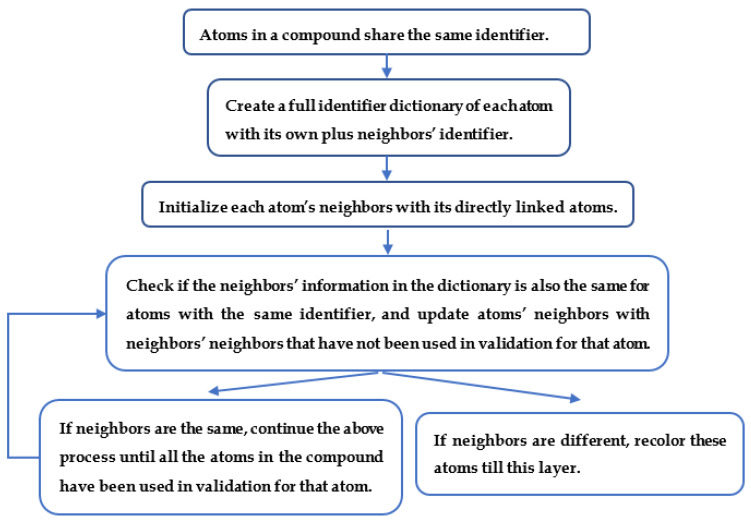
Flow chart of atom coloring.

**Table 1 metabolites-10-00368-t001:** KEGG and MetaCyc databases summary.

Data Types	KEGG	MetaCyc	MetaCyc/KEGG ^a^
Compounds	18636	20264	1.09
Reactions	11427	17203	1.51
Atom-resolved reactions	10282	15909	1.53

^a^ Ratio of MetaCyc entries to Kyoto Encyclopedia of Genes and Genomes (KEGG) entries.

**Table 2 metabolites-10-00368-t002:** Correspondences between KEGG and MetaCyc compounds.

Identifiers	KEGG	MetaCyc	Correspondences
InChI	13216 (70.9%)	15076 (74.4%)	2336
ChEBI	15353 (82.4%)	8404 (41.5%)	3106
KEGG	18636 (100%)	5402 (26.7%)	5402
Either-ID	18636 (100%)	15216 (75.1%)	5681

InChI: IUPAC International Chemical Identifier; ChEBI: Chemical Entities of Biological Interest.

**Table 3 metabolites-10-00368-t003:** Incomplete detection of aromatic substructures by BASS and Indigo.

Databases	BASS	Indigo
KEGG	0	~1500
MetaCyc	30	~1700

**Table 4 metabolites-10-00368-t004:** Compounds with aromatic substructure.

Databases	Count
KEGG	9204 (49.4%)
MetaCyc	8292 (40.9%)

**Table 5 metabolites-10-00368-t005:** Matched compounds detected by the compound coloring identifiers.

Identifiers	Color Matched Pairs	ID Verified Pairs
Tight coloring identifier	1763	1448
Loose coloring identifier	8865	5451

**Table 6 metabolites-10-00368-t006:** Analysis of Enzyme Commission (EC) types involved in reactions in KEGG and MetaCyc.

EC Types	KEGG (Count/Percentage)	MetaCyc (Count/Percentage)
No EC	1263/11.05%	3427/19.92%
1-leveled EC	24/0.21%	11/0.06%
2-leveled EC	126/1.10%	67/0.39%
3-leveled EC	1081/9.46%	2958/17.19%
4-leveled EC	8933/78.17%	10740/62.43%

**Table 7 metabolites-10-00368-t007:** Correspondences between KEGG and MetaCyc compounds verified by reactions.

Conditions	ID-Confirmed Pairs	Possible Pairs
Pairs not in reaction	1224	1122
Pairs in reactions	4227	2292
Verified by 3-leveled EC	3810	1848
Verified by 4-leveled EC	3580	1540

**Table 8 metabolites-10-00368-t008:** Compounds with the same coloring identifier, excluding R groups.

Databases	Tight Coloring Identifier	Loose Coloring Identifier
KEGG	99 (0.5%)	968 (4.8%)
MetaCyc	117 (0.6%)	1144 (5.6%)

**Table 9 metabolites-10-00368-t009:** Compounds gaining asymmetry after addition of extra information in the atom naming.

Databases	Atom Stereochemistry	Atom Charge	Bond Stereochemistry
KEGG	232	38	169
MetaCyc	219	148	227

## Data Availability

All data used and the results generated in this manuscript are available on: https://doi.org/10.6084/m9.figshare.12894008.v1.

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
