# Peer review of "Atom Identifiers Generated by a Neighborhood-Specific Graph Coloring Method Enable Compound Harmonization across Metabolic Databases"

_metabolites, 2020, doi:10.3390/metabo10090368_

Round 1

Reviewer 1 Report

Dear authors,

here are my comments:

General comments:

  • Not all abbreviations are defined – please improve!
  • The manuscript contains numerous spelling mistakes, basically everywhere but especially in the methods part; only some of them are listed in the comments below.
  • Figure and table legend descriptions have to be improved and they should be much more informative!
  • I do not get the point about “graph coloring method” – you nowhere use a coloring method, you give numbers for every atom and not color them. In metabolic networks, graph coloring is a method to assign labels to elements of a graph to subject certain constraints – which is not the case here. Therefore I think you have to rename your method as well as your title
  • I do think that you should much more present your actual results of your method in your Figures - your output is good but I guess it could be better represented

Line 46: „biotransformation routes“   Biotransformation is up to my knowledge defined as modification of chemical compounds to make them from non-excretable to excretable substances. Metabolic networks are rather defined as a set of metabolic, physical and biochemical processes of a cell which, in more detail, describe relationships of metabolites and enzymes that interact together to catalyze biochemical reactions.

Line 49: Please give some examples from other organisms where it does exist – e.g. E. coli, and suggest that conserved pathways can be compared between organisms

Line 57ff: As it is not a review, the introduction part contains already results (e.g. Table 1 and Table 2) – please shift this towards results part;

Line 58: 8.7%? – I do come to another result according to your Table 1 - is it not 8.03% in compounds;

Line 58: Would it not be better to compare how many reactions of total reactions are resolved on the atom-level? So 100% of MetaCyc reactions and “just” 87,6% in KEGG? And again, I do not come to the result of 72% when comparing MetaCyc vs. KEGG.

Line 74: Table 2 – define abbreviations of identifiers in table header! I am confused – where is the total number of Meta-Cyc compounds (20264) of Table1 in Table 2 is visible/included?

Line 83: There are methods available which provide in silico atom labeling automatically and reconstruct metabolic networks .g. Reconstruction of biological pathways and metabolic networks from in silico labeled metabolites by Noushin Hadadi, Jasmin Hafner, Keng Cher Soh and Vassily Hatzimanikatis

Line 100: spelling mistake “representation”

Line 105 / Figure 1 – please check description of the figure – “Correspondence” is, in my opinion, not the correct term, as both are not fully in accordance to each other. I guess you rather meant that you have to make sure that they are in correspondence. For this Figure it would also be beneficial, if you could implement the result of your applied changes of single and double bonds within the aromatic substructure to aromatic bond and with which chemical structure of 2,6-dihydroxynicotinate you are coming up with.

Line 110: /Table 3:  How many and which substances/aromatic substructures were compared? Which types of aromatic (sub)structures? Or is this given in Table 4?

Line 122: Please give total number of compared structures

Line 125: / Figure2:  It would also be nice to have the chemical identifier for this compound; additionally you should describe that the red highlighted aromatic substructure could have been not detected by Indigo and give some other examples of substructures.

Line 140 / Figure 3 description: Please add information, that 1&2; 4&5 etc. do have the same atom identifier – it is not colored, it is actually numbered. It would be beneficial if you also would implement the color-code route towards the not-symmetric part and the coloring validation and if necessary, the recoloring is exemplarily shown!

Line 155: specify “other identifiers”

Line 161/Figure 4: Please also include your solution for shown example

Line170/Table 5: Define tight / loose coloring; also give more details in Table-legend – which pairs are built etc.

Line 175/Table 6: Be more specific in the header, other not listed percentage are e.g. lower leveled ECs

Line 178: spelling mistake comfirmed

Line 192 ff: KEGG aromatic types: can you also compare at the end to MetaCyc – similar problems or none?

Line 199: simple heuristic what? approach?

Line 211 ff: What do you suggest to do so, if comparison with Indigo is also not practicable?

Line 217/Fig. 5: Summarize please also in the Figure legend your findings and provide also an example of S-containing rings; highlight non-aromatic substructures and therefore non-aromatic quinoid structures

Line 220: spelling mistake harmization

Line 222: spelling mistake referecing

Line 225/Fig. 6: Please describe in the legend the problem and provide also your solution to the problem

Line 227: Please show results of your “coloring” identifier to show the discrepancy between the compounds from the different databases.

Line 226 ff: How have you determined that those Metabolites are the same? Via the trivial name? or via the chemical name? or because they were interlinked from e.g. in MetaCyc with each other? Have you calculated/compared the Monoisotopic Mass and therefore can judge of the right chemical structure?

Line 237ff: show an example of loosening

Line 242: even if trivial, define “R” group

Line 248/Table 8: How many substances were compared? (similar question for Suppl. Table S1)

Line 263: How are your results after the heuristic methods?

Line 273/Fig. 8: Please show also your results – also improve figure description

Line 274 ff: any suggestions how to overcome those limitations?

Line 274ff: Isn’t there also a resonance problem in Fig.8 KEGG-C04167?

Line 285ff: You have nowhere discussed/explained how your method can be applied by others? Is there a Graphical user interface etc. available?

Line 294: even if already cited above, please refer to your BASS and the Indigo methods

Line 307: I know that you can find some of the data in the data repository, but can you also list not just mismatches, but also matches in a Supplementary file? I guess this is a valuable information, and it is more easy to access from a supplementary table than to dug in your figshare data!

Line 310: can you also list those 1848 pairs?

Line 335: letters are cut in the figure (e.g. “p”;“y”)

Line 324: molfile is misspelled

Line 326: please refer to a source for “molfile” e.g. your reference 11 Dalby et al.

Line 344: spelling mistake “structure”

Line 352: spelling mistake aromatic

Line 355: spelling mistake

Line 392: letters are cut in the figure (e.g. “p”;“y”); this figure could benefit of a graphical representation of every step

Line 397: spelling mistake; even if trivial, you have to define n, a, k ; and also specify if there is a multiplication sign between every bracket and if so what is exemplarily the specific output

Minor Comments:

Some of the Structures in the figures tend to be blurry

Author Response

Reviewer 1:

Issue 1:

Not all abbreviations are defined – please improve!

Response:

We believe we have now defined all abbreviations.

Issue 2:

The manuscript contains numerous spelling mistakes, basically everywhere but especially in the methods part; only some of them are listed in the comments below.

Response:

We apologize for the numerous spelling issues.  We were under a submission deadline and our focus was on the correctness of the results and interpretation.  We have fixed all grammatical issues pointed out by reviewers and what we have found through a careful rereading.

Issue 3:

Figure and table legend descriptions have to be improved and they should be much more informative!

Response:

We have expanded the legend descriptions.

Issue 4:

I do not get the point about “graph coloring method” – you nowhere use a coloring method, you give numbers for every atom and not color them. In metabolic networks, graph coloring is a method to assign labels to elements of a graph to subject certain constraints – which is not the case here. Therefore I think you have to rename your method as well as your title

Response:

This comes from graph theory terminology and refers to giving nodes specific type (i.e. color).  Our method “colors” a node (i.e. atom) based on its local graph neighborhood which includes element and bond type.  This basic approach is used by all best-in-class common subgraph isomorphism detection methods (even if the published method does not describe the method as using graph coloring).  However, the reviewer’s confusion is due to our lack of description of the terminology being used within the manuscript. We sometimes forget that others are not as familiar with certain mathematical or computational areas.  Therefore, we have revised the following description to make it clearer what “neighborhood-specific graph coloring” means within this context:

“In this paper, we developed a novel neighborhood-specific graph coloring method that creates a unique identifier for each atom in a compound by expanding the type (color) of each atom based on its “neighborhood” of atoms (nodes) bonded (edges) to it.”

We have also revised the title as follows:

“Atom Identifiers Generated by a Neighborhood-Specific Graph Coloring Method Enable Compound Harmonization Across Metabolic Databases”

Issue 5:

I do think that you should much more present your actual results of your method in your Figures - your output is good but I guess it could be better represented

Response:

We chose to use tables to represent most of the results from our methods, because it is clearer than trying to use some Venn or Venn-like diagram representation to do so.  We also spent significant effort on deciding what should go into the main text and what should be in supplemental material. 

Issue 6:

Line 46: „biotransformation routes“   Biotransformation is up to my knowledge defined as modification of chemical compounds to make them from non-excretable to excretable substances. Metabolic networks are rather defined as a set of metabolic, physical and biochemical processes of a cell which, in more detail, describe relationships of metabolites and enzymes that interact together to catalyze biochemical reactions.

Response:

This comes down to the definition of the word “biotransformation”.  Here are standard definitions from different sources:

“the alteration of a substance, such as a drug, within the body.” from  https://www.lexico.com/en/definition/biotransformation .

“the chemical modification (or modifications) made by an organism on a chemical compound.” from  https://en.wikipedia.org/wiki/Biotransformation .

We specifically chose the word “biotransformation” because of its common definition, since not all reactions that occur in organisms are catalyzed by enzymes.  Therefore, we stand by our use of the term here.

Issue 7:

Line 49: Please give some examples from other organisms where it does exist – e.g. E. coli, and suggest that conserved pathways can be compared between organisms

Response:

The comparison of conserved (central metabolism) pathways between organisms is unrelated to the point we are trying to make here.  Both KEGG and MetaCyc represent this comparison of conserved pathways; however, there is considerable lack of confidence in these pathways and their completeness for non-model organisms.  But our point was about the completeness of atom-resolved metabolic networks for human metabolism because of its implication of SIRM research of human metabolism.  

Issue 8:

Line 57ff: As it is not a review, the introduction part contains already results (e.g. Table 1 and Table 2) – please shift this towards results part;

Response:

Tables 1 and 2 represent what is available in the KEGG and MetaCyc databases without our methods.  However, we did perform the evaluation that generated Table 1 and 2 results and agree with the reviewer’s suggestion to move these tables.  Therefore, we have moved Tables 1 and 2 to the beginning of the Results section, while keeping our description of the problem being addressed in the Introduction.

Issue 9:

Line 58: 8.7%? – I do come to another result according to your Table 1 - is it not 8.03% in compounds;

Response:

We rewrote this in terms of a multiplier to make it clearer:

“MetaCyc has 1.09 times as many compound entries as KEGG and 1.72 times as many atom-resolved reaction entries.”

Issue 10:

Line 58: Would it not be better to compare how many reactions of total reactions are resolved on the atom-level? So 100% of MetaCyc reactions and “just” 87,6% in KEGG? And again, I do not come to the result of 72% when comparing MetaCyc vs. KEGG.

Response:

We have added a MetaCyc to KEGG ratio column in Table 1. We have added percentages in parentheses two the KEGG and MetaCyc columns in Table 2.

Issue 11:

Line 74: Table 2 – define abbreviations of identifiers in table header! I am confused – where is the total number of Meta-Cyc compounds (20264) of Table1 in Table 2 is visible/included?

Response:

We have added the abbreviations of identifiers to the Table 2 legend.  With the addition of percentages to the KEGG and MetaCyc columns, the totals are not needed given that they are in Table 1.

Issue 12:

Line 83: There are methods available which provide in silico atom labeling automatically and reconstruct metabolic networks .g. Reconstruction of biological pathways and metabolic networks from in silico labeled metabolites by Noushin Hadadi, Jasmin Hafner, Keng Cher Soh and Vassily Hatzimanikatis

Response:

We have added a section that includes this reference and discusses the generation of hypothetical metabolic networks via reconstruction:

“One approach is to reconstruct a hypothetical atom-resolved metabolic network from generalized reaction descriptions that are atom-specific [11]. However, it is unclear the level of validation and curation that such an approach would require to construct a reasonably accurate atom-resolved metabolic network for generating metabolic models usable in the analysis of SIRM datasets. An alternative is to use curated metabolic databases currently available, in particular the Kyoto Encyclopedia of Genes and Genomes (KEGG) and the MetaCyc metabolic pathway database.”

Issue 13:

Line 100: spelling mistake “representation”

Response:

Fixed.

Issue 14:

Line 105 / Figure 1 – please check description of the figure – “Correspondence” is, in my opinion, not the correct term, as both are not fully in accordance to each other. I guess you rather meant that you have to make sure that they are in correspondence. For this Figure it would also be beneficial, if you could implement the result of your applied changes of single and double bonds within the aromatic substructure to aromatic bond and with which chemical structure of 2,6-dihydroxynicotinate you are coming up with.

Response:

We have changed the Figure 1 legend as follows:

“Correspondence between KEGG and MetaCyc compound entries with different molecular representations.”

Also, these representations are those provided by KEGG and MetaCyc and emphasize the problem being solved, which is identifying correspondence between compound entries from the two databases when they have different molecular representations.

Issue 15:

Line 110: /Table 3:  How many and which substances/aromatic substructures were compared? Which types of aromatic (sub)structures? Or is this given in Table 4?

Response:

Table 4 does provide the total number of compound entries with detected aromatic substructures.  All detected compound entries were used in the comparison of BASS and Indigo.  We have tried to make this clearer by the following revisions:

“First, we compared the aromatic substructures derived by these two methods. As shown in Table 3, Indigo appears more conservative than BASS in detecting aromatic substructures, detecting roughly 85% of what the BASS method does.”

Issue 16:

Line 122: Please give total number of compared structures

Response:

We have added percentages of the total number of compounds in each database to Table 4.

Issue 17:

Line 125: / Figure2:  It would also be nice to have the chemical identifier for this compound; additionally you should describe that the red highlighted aromatic substructure could have been not detected by Indigo and give some other examples of substructures.

Response:

This was an oversight on our part.  We have added the compound entry identifier.  However, we describe our hypothesized reason for the Indigo’s failings in the text and the FigShare repo has all of the examples.  We do not see a good reason to provide more examples in Figure 2, since Figure 2 accomplishes its purpose of showing an example that supports our hypothesis.

Issue 18:

Line 140 / Figure 3 description: Please add information, that 1&2; 4&5 etc. do have the same atom identifier – it is not colored, it is actually numbered. It would be beneficial if you also would implement the color-code route towards the not-symmetric part and the coloring validation and if necessary, the recoloring is exemplarily shown!

Response:

We have added the specific atom identifiers being discussed to better illustrate the problem.  Also, “color” in this context refers to atom identifiers that describe element and local graph topology that includes bond type.

Issue 19:

Line 155: specify “other identifiers”

Response:

We have made the following revisions to explicitly describe the identifiers and refer back to Table 2:

“With the relatively specific coloring identifiers, 1762 correspondences between KEGG and MetaCyc compounds can be detected (see Table 5), which is not satisfactory compared to 5681 pairs discovered by other identifiers (e.g. KEGG, CHEBI, and InChI as shown in Table 2).”

Issue 20:

Line 161/Figure 4: Please also include your solution for shown example

Response:

We have added a harmonized representation in between the two database representations.

Issue 21:

Line170/Table 5: Define tight / loose coloring; also give more details in Table-legend – which pairs are built etc.

Response:

We have added an example of tight and loose atom colorings for Figure 4 into the supplementary material due to the length of the atom color identifiers:

“S Figure 1 uses the MetaCyc CPD-20570 and KEGG C13014 as an example, illustrating how loose coloring addresses the issues caused by tight coloring and facilitates compound harmonization.”

Issue 22:

Line 175/Table 6: Be more specific in the header, other not listed percentage are e.g. lower leveled ECs

Response:

Less specific EC designations are not that useful. However, per the advice of the reviewer, we have now included them in Table 6.

Issue 23:

Line 178: spelling mistake comfirmed

Response:

Fixed.

Issue 24:

Line 192 ff: KEGG aromatic types: can you also compare at the end to MetaCyc – similar problems or none?

Response:

MetaCyc does not provide a molecularly descriptive atom type.  So there is nothing to compare.

Issue 25:

Line 199: simple heuristic what? approach?

Response:

The word “heuristic” has several definitions, one of them being “a heuristic process or method.” However, the reviewer’s confusion indicates that we should make the meaning more explicit.  Therefore, we have added the following revision:

“We used a simple heuristic method (i.e a simple deterministic decisioning approach) to consider oxygen and sulfur atoms as aromatic when they are part of a ring where all other carbon and nitrogen atoms are labeled as aromatic, based on KEGG atom types.”

Issue 26:

Line 211 ff: What do you suggest to do so, if comparison with Indigo is also not practicable?

Response:

There is nothing to be done.  We are not proposing a foolproof solution to aromatic substructure detection, since this an unsolved problem in chemoinformatics.  Too many others with vastly superior expertise in aromaticity have failed to create a robust general solution.  Part of the problem is that aromaticity is a continuous property due to the occupation of multiple resonant, tautomeric, and ionization forms.  This also means that aromaticity depends on various properties of the chemical environment like pH.  Since the focus of the manuscript is not aromatic substructure detection, we did not want to delve into the problems of that field.

Issue 27:

Line 217/Fig. 5: Summarize please also in the Figure legend your findings and provide also an example of S-containing rings; highlight non-aromatic substructures and therefore non-aromatic quinoid structures

Response:

We have added the following to the Figure 5 legend:

“The middle ring of compound C03861 (left) is not aromatic while the middle ring of compound C07729 (right) is aromatic.”

However, we do not want to change Figure 5, since it was designed to highlight the problems of incomplete KEGG aromatic atom types.  Instead, we have added Supplemental Figure 2 with S-containing aromatic ring examples, which is mentioned in the main text:

“In addition, S Figure 2 shows a KEGG compound with an S-containing aromatic ring.”

Issue 28:

Line 220: spelling mistake harmization

Response:

Fixed.

Issue 29:

Line 222: spelling mistake referecing

Response:

Fixed.

Issue 30:

Line 225/Fig. 6: Please describe in the legend the problem and provide also your solution to the problem

Response:

We add the following text to the Figure 6 legend:

“These two corresponding compound entries across KEGG and MetaCyc have two different aromatic substructure representations.”

However, the description of the solution is best left in the main text.

Issue 31:

Line 227: Please show results of your “coloring” identifier to show the discrepancy between the compounds from the different databases.

Response:

We already did this for Figure 4 with the addition of S Figure 1.  Also, these molecules have a lot of atom identifiers and would not fit well into this figure.

Issue 32:

Line 226 ff: How have you determined that those Metabolites are the same? Via the trivial name? or via the chemical name? or because they were interlinked from e.g. in MetaCyc with each other? Have you calculated/compared the Monoisotopic Mass and therefore can judge of the right chemical structure?

Response:

The MetaCyc compound entry has a direct reference to the KEGG compound entry.  This is clearly stated in the text, but we have revised it to make this point clearer:

“Compound CPD-19437 in MetaCyc has a direct reference to KEGG compound C12187, but their coloring identifiers are different (see Figure 7).”

Also, the comparison of the monoisotopic mass is not a good method for compound harmonization.  For one, this is not experimental data.  Two, isomers are going to have the same monoisotopic mass, since the entries will have the same chemical formula. Please refer to our previously published work for an understanding of the level of isomerization in KEGG: Joshua M. Mitchell, Teresa, W-M. Fan, Andrew N. Lane, Hunter N.B. Moseley. "Development and In silico Evaluation of Large-Scale Metabolite Identification Methods using Functional Group Detection for Metabolomics" Frontiers in Genetics - Systems Biology 5, 237 (2014). http://dx.doi.org/10.3389/fgene.2014.00237

Issue 33:

Line 237ff: show an example of loosening

Response:

Figure 4 now has an example. Therefore, we mentioned the Figure 4 example in this section:

“During the compound harmonization process, tight atom and compound coloring was loosened (see Figure 4 for an example)…”

Issue 34:

Line 242: even if trivial, define “R” group

Response:

We have added a basic R group definition as follows:

“Here, we did not count compounds with a generic R group representing ambiguous functional groups and substructures; however, the results that include all compounds are described in Supplementary Table 1.”

Issue 35:

Line 248/Table 8: How many substances were compared? (similar question for Suppl. Table S1)

Response:

We have added percentages to Table 8 and Supplemental Table 1, as we have done for other tables.

Issue 36:

Line 263: How are your results after the heuristic methods?

Response:

We were mentioning this as a future possibility and not something we have done.  We have modified the main text to make this point clearer:

“A heuristic method could be used to test if symmetric atoms are affected by bond and atom stereochemistry, and then atom coloring identifiers incorporated with bond and atom stereochemistry will be generated to overcome this issue.” 

Issue 37:

Line 273/Fig. 8: Please show also your results – also improve figure description

Response:

We have improved the Figure 8 legend and figure.

Issue 38:

Line 274 ff: any suggestions how to overcome those limitations?

Response:

None that we are able to share at this time.  It cannot be done by a simple comparison of compound representations.  The ideas we have need to be investigated and vetted first.

Issue 39:

Line 274ff: Isn’t there also a resonance problem in Fig.8 KEGG-C04167?

Response:

We do not see a resonance problem with KEGG C04167.

Issue 40:

Line 285ff: You have nowhere discussed/explained how your method can be applied by others? Is there a Graphical user interface etc. available?

Response:

This is methods development and not production level software.  This is why all code and results from this manuscript are in a FigShare repository and not on GitHub.  However, we have many production level software packages available on GitHub, PyPI, CRAN, and Bioconductor: https://github.com/MoseleyBioinformaticsLab .

Issue 41:

Line 294: even if already cited above, please refer to your BASS and the Indigo methods

Response:

We have added these references in the Discussion section.

Issue 42:

Line 307: I know that you can find some of the data in the data repository, but can you also list not just mismatches, but also matches in a Supplementary file? I guess this is a valuable information, and it is more easy to access from a supplementary table than to dug in your figshare data!

Response:

We have added this to the supplemental material and mentioned it in the main text:

“About 8865 correspondences between KEGG and MetaCyc are detected (see Table 5 and spreadsheets in supplementary material), and 5451 of them can be confirmed by other identifiers.”

Issue 43:

Line 310: can you also list those 1848 pairs?

Response:

This is included as a column in the supplemental table of matches.

Issue 44:

Line 335: letters are cut in the figure (e.g. “p”;“y”)

Response:

We see no problem with Figure 11. May be issues caused by submission and conversion for review.

Issue 45:

Line 324: molfile is misspelled

Response:

Fixed.

Issue 46:

Line 326: please refer to a source for “molfile” e.g. your reference 11 Dalby et al.

Response:

We added the reference after “molfile bond designation 4”.

Issue 47:

Line 344: spelling mistake “structure”

Response:

Fixed.

Issue 48:

Line 352: spelling mistake aromatic

Response:

Fixed.

Issue 49:

Line 355: spelling mistake

Response:

Fixed.

Issue 50:

Line 392: letters are cut in the figure (e.g. “p”;“y”); this figure could benefit of a graphical representation of every step

Response:

Again, we see no issue with Figure 12. This is likely a artifact of the submission and conversion process of the journal.

Issue 51:

Line 397: spelling mistake; even if trivial, you have to define n, a, k ; and also specify if there is a multiplication sign between every bracket and if so what is exemplarily the specific output

Response:

The formula represents string concatenation.  We have added variable definitions and revised description:

“The formulation is shown in Equation 1, which represents the order of string concatenation with nk being the number of atoms with coloring ak.  The parenthesis and bracket characters are included in the resulting string.”

Issue 52:

Minor Comments:

Some of the Structures in the figures tend to be blurry

Response:

Like with Figures 11 and 12, this may be an artifact of the submission and conversion process.  We will keep an eye on this and regenerate the figures at a higher resolution if possible.

Reviewer 2 Report

Although the problem of database harmonization that the authors try to tackle with this manuscript is an interesting topic, I find the manuscript is still not mature enough for publication. I would like to encourage the authors conducting the following amendments/improvements to the manuscript:

  1. The work will benefit from a third database harmonization to prove that can be generalizable. I miss mentioning possible databases containing similar information to KEGG and MetaCyc in the discussion, can this approach be used on them? Yes/No? why?
  2. The results section present some figures to illustrate examples of the problems encountered and some basic statistics but fail to present an example of what do you gain by this harmonization? I would like to see a pathway where the integration of common identifiers lead to gain in insight.
  3. I think the motivation for similar integrative approaches is not strength enough in the introduction section.

Style/Layout/Grammar

I found several typos during my read and problems with the figures and caption, so I would recommend the authors double check the manuscript.

  • Figures showing molecules from different databases contain screenshots of the same/similar molecule with different colors and it is confusing. Why not using the same layout/colors for all molecules for the paper instead of an screenshot? Draw them with the same software please
  • the abstract contain synonyms like what is more, furthermore, moreover and in addition. I would improve the flow of the abstract by replacing them with other sentences that do not have the similar structure.
  • In line 51, the authors bring to the reader "molfile" without explaining what it is in detail, I feel it is an important format since the authors use it in their analysis that has to be introduced more in detail.
  • The authors mention KEGG RPAIR that is discontinued since 2016 (https://www.genome.jp/kegg/kegg1a.html) how did the author get the information from KEGG, a resource that is not easy to extract information from? I miss that in the supplement (which it wasnt included in the manuscript, i dont know why) -> e.g., did you query the KEGG API?
  • Table 1 stats are generated based on your analysis on are they reported in the original resources?
  • I would like to see a Venn Diagram of the overlap for each entity (enzymes and chemicals)
  • Figure 1, 3 and other seem to be cut on the bottom part of the image.

Typos:

  • line 219: respresentations
  • line 100: represnetation
  • line 119: comfirmed
  • line 120: what does supplemented mean? enriched?
  • line 121: improve english
  • line 177: sifted (filter?)
  • line 179: furuer
  • line 219: respresentations
  • line 241: don't
  • line 311: identifies
  • line 324: mofile
  • line 312: in version May ... (grammar)
  • line 336: mofile
  • line 360: hydrgen

  • Caption (Figure 12)
  • - Example

Author Response

Reviewer 2:

Although the problem of database harmonization that the authors try to tackle with this manuscript is an interesting topic, I find the manuscript is still not mature enough for publication. I would like to encourage the authors conducting the following amendments/improvements to the manuscript:

Response:

We were under a deadline to submit by a certain date.  The grammatical issues resulted from our focus on making sure the results and interpretations were correct.

Issue 1:

The work will benefit from a third database harmonization to prove that can be generalizable. I miss mentioning possible databases containing similar information to KEGG and MetaCyc in the discussion, can this approach be used on them? Yes/No? why?

Response:

We demonstrate that this approach is an improvement on prior methods, mostly provided by the databases themselves.  We must push back against the reviewer’s request that we demonstrate this on three databases.  It is straightforward to see that three sets of molecular identifiers built from atom coloring identifiers can be compared as easily as two sets.

Also, we clearly describe the requirements for our compound harmonization method in the Discussion section:

“Furthermore, the graph coloring method and compound harmonization approach can be used to integrate any metabolic database that provides a molfile representation of compounds, greatly facilitating future construction of more complete integrated metabolic networks.”

Issue 2:

The results section present some figures to illustrate examples of the problems encountered and some basic statistics but fail to present an example of what do you gain by this harmonization? I would like to see a pathway where the integration of common identifiers lead to gain in insight.

I think the motivation for similar integrative approaches is not strength enough in the introduction section.

Response:

Compound harmonization has several uses including combining analyses derived from both KEGG and MetaCyc as well as the integration of their metabolic networks at an atom-resolved level.  We have mentioned some of this motivation in the Discussion section:

“…, greatly facilitating future construction of more complete integrated metabolic networks.”

However, we do see the reviewer’s perspective about making these points in the introduction.  Towards this end, we have revised the following to the introduction:

“To our knowledge, this is the first attempt to create unique atom and compound identifiers that are consistent with respect to molecular symmetry and for the explicit purpose of harmonizing compounds across the KEGG and MetaCyc databases, ultimately to facilitate the construction of an integrated atom-resolved metabolic network.”

Issue 3:

Style/Layout/Grammar

I found several typos during my read and problems with the figures and caption, so I would recommend the authors double check the manuscript.

Response:

We apologize for the number of grammatical mistakes, but we were rushed.  We have double checked the manuscript.

Issue 4:

Figures showing molecules from different databases contain screenshots of the same/similar molecule with different colors and it is confusing. Why not using the same layout/colors for all molecules for the paper instead of an screenshot? Draw them with the same software please

the abstract contain synonyms like what is more, furthermore, moreover and in addition. I would improve the flow of the abstract by replacing them with other sentences that do not have the similar structure.

Response:

Many of these representations come from the databases themselves.  And the fact that they are different helps to emphasize the problem we are solving: matching compounds that have different molecular representations across the two databases.

Issue 5:

In line 51, the authors bring to the reader "molfile" without explaining what it is in detail, I feel it is an important format since the authors use it in their analysis that has to be introduced more in detail.

Response:

This is a valid point.  We sometimes forget that not everyone is familiar with standard chemoinformatics concepts and knowledge.  We have added a brief description of what a molfile is:

“The popular molfile description of a compound is a text-based chemical table file format developed by MDL Information Systems and contains information about atoms, bonds, connectivity, and coordinates [12], which is available in most databases.”

Issue 6:

The authors mention KEGG RPAIR that is discontinued since 2016 (https://www.genome.jp/kegg/kegg1a.html) how did the author get the information from KEGG, a resource that is not easy to extract information from? I miss that in the supplement (which it wasnt included in the manuscript, i dont know why) -> e.g., did you query the KEGG API?

Response:

The KEGG RPAIR database was not used in this work; therefore, we have not mentioned it further.  We are very adept at querying KEGG and several other major scientific databases and repositories, which we have published methods for these purposes.  However, KEGG has some restrictions on its use.  One should be careful not to send too many repetitive requests to their REST API, which is true for most major scientific repositories.

Issue 7:

Table 1 stats are generated based on your analysis on are they reported in the original resources?

I would like to see a Venn Diagram of the overlap for each entity (enzymes and chemicals)

Response:

All of the results were generated from our analyses based on the full database pull we had performed on the appropriate sections of both KEGG and MetaCyc.  We deemed Venn diagrams as not a good choice given the complexity of some of the results.  Also, we have not analyzed enzymes and reactions for harmonization in this work.  That is planned for a future project.

Issue 7:

Figure 1, 3 and other seem to be cut on the bottom part of the image.

Response:

This is likely an artifact of the journal submission and conversion process.  Another reviewer mentioned letters being cut off of the side.  All of the figures look fine in the word document we submitted.

Issue 8:

Typos:

line 219: respresentations

line 100: represnetation

line 119: confirmed

Response:

We have fixed all of these spelling mistakes. 

Issue 9:

line 120: what does supplemented mean? enriched?

Response:

We do not understand the problem with the use of the word “supplemented” on line 120.  However, we did make a minor change to sentence to make another part clearer:

“Therefore, we supplemented the KEGG aromatic substructures with additional Indigo-detected substructures from MetaCyc.”

Issue 10:

line 121: improve English

Response:

We rewrote the sentence as follows:

“By using both methods, we were able to detect aromatic substructures in about half of the compounds in each database.”

Issue 10:

line 177: sifted (filter?)

Response:

We rewrote the sentence as follows:

“We first identified color-harmonized pairs that both take part in some reactions in their respective database.”

Issue 11:

line 179: furuer

line 219: respresentations

line 241: don't

line 311: identifies

line 324: mofile

line 312: in version May ... (grammar)

line 336: mofile

line 360: hydrgen

Response:

We have fixed all of these spelling mistakes.  We have also proof-read the manuscript again.

Issue 12:

Caption (Figure 12)

- Example

Response:

We do not know what the reviewer was trying to point out here.

Reviewer 3 Report

As stated in the abstract, the authors have “developed a graph coloring method that creates unique identifiers for each atom in a compound facilitating construction of an atom-resolved metabolic network”. It is suggested that their “method is guaranteed to generate the same identifier for  symmetric atoms, enabling automatic identification of possible additional mappings caused by  molecular symmetry”. 

The work presented is much needed research as well as necessary technology. Many efforts over the years have tackled various aspects of this problem. 

Major: 

In the manuscript, the terms molecular symmetry and atom symmetry and substructure are used but their scope is never clearly defined. Are we considering rigid molecules, flexible molecules or both? What kind of molecules are in the category of "flexible", if considered? Are the authors considering graph symmetries or 3D symmetries? (what is the group of symmetries considered?) If the authors consider flexible molecules, can they provide an example of how the protons on CH3 with degenerate symmetries are labeled? And, please elaborate on consideration of a threshold under which two flexible and symmetric structures are considered identical (or different)? If the consideration is limited to rigid molecules only, then it should be stated clearly. 

The following sentence in the manuscript is confusing: “a compound coloring identifier derived from the corresponding atom coloring identifiers can be used for compound harmonization across various metabolic network databases, which is an essential first step in network integration.“  Is this statement addressing a particular class of molecules, or particular colorings that include specific atoms, or symmetries. For example,
The manuscript evaluated the overlap between KEGG and MetaCyC in the introduction in order to make the case.  Using InChI to perform this task is known to be error prone (according to author’s reference 18), and software for correcting these IDs based on InChI has been provided in the literature (according to author’s reference 19). Moreover, references 18 and 19 of the author report on a complete and corrected database with unique IDs and unique labeling that includes protons. It is not clear why the author’s did not use a reliable reference database of IDs to perform this overlap test since the problem had already been discussed and solved in the literature. If this statement is referring to lack of support for representation of symmetries only, then this statement needs to be clarified and the examples should focus on this only. 

In light of references 18 and 19, It is unclear what the following statement in the manuscript indicates: No appropriate method has been previously published that provides each atom in a compound a useful identifier for constructing an atom-resolved metabolic network.   Can the authors comment on this and provide examples?  In what sense the existing unique labelings do not address useful identifiers. Is it specifically related to symmetries, or is it true in a more broad sense? If symmetries, the manuscript should state that their method is addressing the lack of accounting for symmetries in current identifiers.  As stated, it seems that the authors believe that existing identifiers do not uniquely label all atoms. If so, then examples are appropriate. 

In the results section, the authors begin with the comparison of CPD-6962 in MetaCyc and KEGG referencing compound C15523 (Figure 1) and suggest that they have different labelings. To me, this seems to be a problem with MetaCyc and KEGG not using unique identifiers.  Based on what I was able to find out, these two compounds have been given the same InChI identifier by the approach of reference 18.  Perhaps a more correct statement is that these databases are not using unique IDs. It is also correct to say that current IDs do not identify and record symmetries, but existing IDs do recognize the molecules to be the same. 

A substructure search is considered a very hard problem in computer science (subgraph isomorphism), and if symmetries are added, this problem becomes yet a harder problem.  The manuscript presents a computational experiment, but it does not elaborate on the limitations of the approximate algorithm for doing this search. (I assume that it is not an exact algorithm because of the hardness of the problem). It would be helpful if the manuscript discusses how the approximate search result could potentially impact their overall aim of the manuscript (harmonization).

On lines 154-155, “With the relatively specific 154  coloring identifiers, 1762 correspondences between KEGG and MetaCyc compounds can be detected 155  (see Table 5), which is not satisfactory compared to 5681 pairs discovered by other identifiers.  Did the authors verify this with a unique labeling approach - for example the method used in 18? 

It is unclear as to how the representations will be used in practice in metabolic networks (author's goal in the abstract).  For example, if it is required that isotopic modifications be kept track of in a pathway, the set of possible transformation in each step must be tracked.  In a long chain, the number of possible combinations grows, but so does the number of potentially identical states.  How does the model presented can help keep track of these?  How are the presentations mapped?  Would the authors consider mappings that account for kinetic isotope effects?  More explanation in the discussion would be useful.

Overall, the proposed approach seems promising, but the readers would benefit from a refined scope of application and clarifications regarding the specific unsolved problem that is being addressed. 

Author Response

Reviewer 3:

As stated in the abstract, the authors have “developed a graph coloring method that creates unique identifiers for each atom in a compound facilitating construction of an atom-resolved metabolic network”. It is suggested that their “method is guaranteed to generate the same identifier for  symmetric atoms, enabling automatic identification of possible additional mappings caused by  molecular symmetry”.

The work presented is much needed research as well as necessary technology. Many efforts over the years have tackled various aspects of this problem.

Response:

We thank the reviewer for recognizing the value of the research we present.

Issue 1:

Major:

In the manuscript, the terms molecular symmetry and atom symmetry and substructure are used but their scope is never clearly defined. Are we considering rigid molecules, flexible molecules or both? What kind of molecules are in the category of "flexible", if considered? Are the authors considering graph symmetries or 3D symmetries? (what is the group of symmetries considered?) If the authors consider flexible molecules, can they provide an example of how the protons on CH3 with degenerate symmetries are labeled? And, please elaborate on consideration of a threshold under which two flexible and symmetric structures are considered identical (or different)? If the consideration is limited to rigid molecules only, then it should be stated clearly.

Response:

We are only considering molecular configuration (i.e. changes requiring the breaking of a bond) and not conformation (i.e. changes not requiring the breaking of a bond like a bond rotation) in the tight atom coloring.  We have added the following sentence to make this point clear:

“In this context, only molecular configuration (i.e. changes requiring the breaking of a bond) and not molecular conformation (i.e. changes not requiring the breaking of a bond like a bond rotation) are considered in the generation of these identifiers.”

Issue 2:

The following sentence in the manuscript is confusing: “a compound coloring identifier derived from the corresponding atom coloring identifiers can be used for compound harmonization across various metabolic network databases, which is an essential first step in network integration.“  Is this statement addressing a particular class of molecules, or particular colorings that include specific atoms, or symmetries. For example,

The manuscript evaluated the overlap between KEGG and MetaCyC in the introduction in order to make the case.  Using InChI to perform this task is known to be error prone (according to author’s reference 18), and software for correcting these IDs based on InChI has been provided in the literature (according to author’s reference 19). Moreover, references 18 and 19 of the author report on a complete and corrected database with unique IDs and unique labeling that includes protons. It is not clear why the author’s did not use a reliable reference database of IDs to perform this overlap test since the problem had already been discussed and solved in the literature. If this statement is referring to lack of support for representation of symmetries only, then this statement needs to be clarified and the examples should focus on this only.

Response:

This sentence is focused on the problem of integrating metabolic network databases like KEGG and MetaCyc.  A major first step in metabolic network integration is matching equivalent compounds (metabolites) across both databases.  Our work demonstrates an approach that builds a unique compound identifier from atom coloring identifiers based on local molecular graph topology.

The atom coloring method presented generates unique identifiers that are the same for symmetric atoms.  These are very useful for the generation of an atom-resolved metabolic network, since these identifiers would simplify the generation of all relevant atom mappings across each reaction.  Moreover, the atom coloring method allows for a controlled loosening of the coloring criteria in order to find correspondent compound entries with different molecular representations.

Reference 18 from Dr. Eghbalnia (and his colleagues) is focused on an evaluation of PubChem consistency and not cross-database harmonization, especially when the molecular representations are inconsistently represented across databases.  Reference 19 is focused on creating unique atom identifiers for small (organic) molecules, addressing a problem in the InChI Trust software not generating a consistent ordering of hydrogen atoms from an InChI string. To our understanding, this solution is rather computationally intensive. While reference 19 from Dr. Eghbalnia do create unique identifiers for atoms, the methods from both papers are not directly useful for compound harmonization across KEGG and MetaCyc, especially since their approach cannot handle molfiles with R-groups.  Our approach improves compound harmonization and creates consistent atom identifiers for symmetric atoms, while dealing with pragmatic issues arising from differing molecular representations being used in KEGG and MetaCyc.  

Issue 3:

In light of references 18 and 19, It is unclear what the following statement in the manuscript indicates: No appropriate method has been previously published that provides each atom in a compound a useful identifier for constructing an atom-resolved metabolic network.   Can the authors comment on this and provide examples?  In what sense the existing unique labelings do not address useful identifiers. Is it specifically related to symmetries, or is it true in a more broad sense? If symmetries, the manuscript should state that their method is addressing the lack of accounting for symmetries in current identifiers.  As stated, it seems that the authors believe that existing identifiers do not uniquely label all atoms. If so, then examples are appropriate.

Response:

While the method in reference 19 (ALATIS) could generate unique identifiers, these identifiers are not consistent for symmetric atoms and thus not useful for the construction of an atom-resolved metabolic network. Also, ALATIS cannot handle molfiles with R groups because the InChI Trust software cannot handle molfiles with R groups.

We have tried to highlight these points in the manuscript:

“One group tried to assign a unique name for every atom in the compound based on the compound’s International Union of Pure and Applied Chemistry (IUPAC) International Chemical Identifier (InChI) representation [19], which does not apply to this scenario since symmetric atoms can share the same routes in the metabolic network. Also, any InChI-based approach cannot handle the compound entries with R-groups.  To our knowledge, no appropriate method has been previously published that provides each atom in a compound with a useful identifier for the explicit purpose of constructing an atom-resolved metabolic network, either because the identifier was not unique or because it was not consistent for symmetric atoms.”

Issue 4:

In the results section, the authors begin with the comparison of CPD-6962 in MetaCyc and KEGG referencing compound C15523 (Figure 1) and suggest that they have different labelings. To me, this seems to be a problem with MetaCyc and KEGG not using unique identifiers.  Based on what I was able to find out, these two compounds have been given the same InChI identifier by the approach of reference 18.  Perhaps a more correct statement is that these databases are not using unique IDs. It is also correct to say that current IDs do not identify and record symmetries, but existing IDs do recognize the molecules to be the same.

Response:

Yes, both databases use different molecular representations for the same chemical entity.

Molecular identifiers like InChI do help to identify the same compound in two different databases, but only when the molecular representations generate the same InChI.  Table 2 shows what can be corresponded using the molecular identifiers provided by the two databases.  Also, not all molfile representations provided by both databases easily generate InChI from the InChI Foundation software. 

InChI does not provide unique identifiers for specific atoms.

Issue 5:

A substructure search is considered a very hard problem in computer science (subgraph isomorphism), and if symmetries are added, this problem becomes yet a harder problem.  The manuscript presents a computational experiment, but it does not elaborate on the limitations of the approximate algorithm for doing this search. (I assume that it is not an exact algorithm because of the hardness of the problem). It would be helpful if the manuscript discusses how the approximate search result could potentially impact their overall aim of the manuscript (harmonization).

Response:

We know it is a very hard problem.  Our initial solution to this problem was published back in 2014:

Joshua M. Mitchell, Teresa, W-M. Fan, Andrew N. Lane, Hunter N.B. Moseley. "Development and In silico Evaluation of Large-Scale Metabolite Identification Methods using Functional Group Detection for Metabolomics" Frontiers in Genetics - Systems Biology 5, 237 (2014).

We have made improvements in our algorithm, which are used in this work; however, this manuscript is not the appropriate place to describe this problem and the solution we have developed, especially since we reference our prior work in this manuscript (the BASS reference).  However, we have elaborated further on why the algorithm performs well, which is due to its use of neighborhood-specific graph coloring. 

“Two independent aromatic detection methods were used in aromatic substructure detection: our BASS substructure detection method [21] which uses neighborhood-specific graph coloring [22] to greatly improve subgraph isomorphism detection [23]…”

But in our opinion, the manuscript has enough to present and discuss, without describing another highly computational problem at the level of detail suggested by the reviewer.  However, we are working on a new code base with an API and CLI along with a manuscript that would describe the set of related common subgraph isomorphism problems we can solve now with rather good computational performance.  It is an exact solution and is only practical for smaller graphs like molecular graphs.  It would likely fail to finish in reasonable time if the subgraph size exceeds a few thousand atoms.

Also, we already describe how the common subgraph isomorphism analysis impacts harmonization via the aromatic substructure search.  We have emphasized this in the manuscript as follows:

“We used two methods in aromatic substructure detection. One is based on common subgraph isomorphism detection … Then, we built a set of all aromatic substructures detected from the KEGG compounds without duplication. Furthermore, we manually inspected the set of aromatic substructures to ensure data quality. With this curated set of reference aromatic substructures, we tested each compound in a database for the presence of any of these aromatic substructures using the BASS method [21].”

Issue 6:

On lines 154-155, “With the relatively specific 154  coloring identifiers, 1762 correspondences between KEGG and MetaCyc compounds can be detected 155  (see Table 5), which is not satisfactory compared to 5681 pairs discovered by other identifiers.  Did the authors verify this with a unique labeling approach - for example the method used in 18?

Response:

We implemented a recursive local topology search algorithm in order to validate that the atom identifiers generated were consistent for symmetric atoms.  Once the atom identifiers were validated, we used database-provided molecular IDs to validate our molecular IDs generated from the concatenation of the unique atom identifiers.  This was followed up with manual inspection of the few cases where our correspondence pairs did not match the database-provided molecular IDs.   We have added this to the manuscript:

“With both tight and loose compound coloring identifiers, about 95.95% compounds pairs detected by other chemical IDs can be discovered. We manually checked the compound pairs that were discordant with other chemical IDs and found that none of them are caused by an inconsistency between the coloring identifier and the compound representation.”

Issue 7:

It is unclear as to how the representations will be used in practice in metabolic networks (author's goal in the abstract).  For example, if it is required that isotopic modifications be kept track of in a pathway, the set of possible transformation in each step must be tracked.  In a long chain, the number of possible combinations grows, but so does the number of potentially identical states.  How does the model presented can help keep track of these?  How are the presentations mapped?  Would the authors consider mappings that account for kinetic isotope effects?  More explanation in the discussion would be useful.

Response:

This manuscript only deals with the quick generation of unique atom identifiers and compound identifiers and their use in compound harmonization.  We have not presented a solution yet to atom-resolved metabolic network integration, but both the generation of unique atom identifiers that are consistent for symmetric atoms as well as compound harmonization are necessary steps towards this goal.  We are working on atom-resolved metabolic network integration next. 

Issue 8:

Overall, the proposed approach seems promising, but the readers would benefit from a refined scope of application and clarifications regarding the specific unsolved problem that is being addressed.

Response:

We thank the reviewer for their positive respond and have tried to improve the manuscript description of the problem being solved and the solutions presented.

Round 2

Reviewer 2 Report

The authors have addressed all my questions

Author Response

Reviewer 2:

The authors have addressed all my questions

Response:

We thank the reviewer for their effort in reviewing our manuscript!

Reviewer 3 Report

The authors have clarified most of my concerns.  One major remaining concern  and some suggestions and comments follow.

A major concern is the presentation of table 1.  It has already been established that many databases have incorrect InChI strings.  Because of this, it is unclear if all or some of the results in the table are due to incorrect InChI strings. Since corrected InChI strings have ben reported (ALATIS referenced by the authors), the data in the table should report the mismatches using these corrected InChI. This will support the important point raised by authors.  Otherwise, the issue of incorrect InChI is confounded with the issues that the authors are addressing. With this correction addressed, a few minor issues should also be addressed.

First, the mathematical and computational challenges of molecular graph theory has been captured in a recent book by Erica Flapan (When Topology Meets Chemistry. A Topological Look at Molecular Chirality by Erica Flapan).  The rigorous language in this book goes a long way toward clarifying the issues of symmetry.  For example, the rigorous definition of symmetry is given - in simple terms:

point groups < molecular symmetry group < topological symmetry group < automorphism group (< means contained in)

Point groups: automorphisms induced by rotations and reflections 

Molecular symmetry groups: automorphisms induced by molecular motions 

Topological symmetry groups: automorphisms induced by homeomorphisms of space 

Automorphism groups: automorphisms of the abstract graph 

Second, the following three references should be added:

Methods and algorithms III. Graph invariants and stabilization methods

G. Tinhofer and M. Klin, “Algebraic combinatorics in mathematical chemistry. Tech. Rep. TUM -M9902, Technische Universitat Munchen, 1999.

Get Your Atoms in Order—An Open-Source Implementation of a Novel and Robust Molecular Canonicalization Algorithm

Nadine Schneider, Roger A. Sayle, and Gregory A. Landrum

Journal of Chemical Information and Modeling 2015 55 (10), 2111-2120

DOI: 10.1021/acs.jcim.5b00543

ArbAlign: A Tool for Optimal Alignment of Arbitrarily Ordered Isomers Using the Kuhn–Munkres Algorithm

Berhane Temelso, Joel M. Mabey, Toshiro Kubota, Nana Appiah-Padi, and George C. Shields

Journal of Chemical Information and Modeling 2017 57 (5), 1045-1054

DOI: 10.1021/acs.jcim.6b00546

Finally, with respect to the analysis of metabolic pathways and issues related to graph theory, the joint work of IUBMB and IUPAC should be acknowledged

Author Response

Reviewer 3:

The authors have clarified most of my concerns.  One major remaining concern  and some suggestions and comments follow.

Response:

We worked diligently to address all of the concerns raised in the first round.

Issue 1:

A major concern is the presentation of table 1.  It has already been established that many databases have incorrect InChI strings.  Because of this, it is unclear if all or some of the results in the table are due to incorrect InChI strings. Since corrected InChI strings have ben reported (ALATIS referenced by the authors), the data in the table should report the mismatches using these corrected InChI. This will support the important point raised by authors.  Otherwise, the issue of incorrect InChI is confounded with the issues that the authors are addressing. With this correction addressed, a few minor issues should also be addressed.

Response:

Table 1 deals with the number of compound reaction entries in each database.  We assume that the reviewer is referring to Table 2 which deals with several compound identifiers including InChI.  However, we feel that the reviewer is making a too strong of an argument that “all or some of the results in the table are due to incorrect InChI strings.” The reviewer is implying that the majority of InChI strings are wrong in one or both databases.  This is highly unlikely as the ATLATIS publication on InChI errors in PubChem indicate that only a minor fraction of PubChem database entries has such errors.  Moreover, the presence of such errors would further support the use of the compound harmonization methods presented in this manuscript.  Also, Table 2 deals with two other compound identifiers, ChEBI and KEGG Compound IDs.  Therefore, we see the reviewer’s comment as ill-conceived and mute.

However, there is another related issue not mentioned by the reviewer.  What compound harmonization is possible if InChI were uniformly generated for all database compound entries?  This issue, though not raised by reviewer, is very valid in this context.  Towards this end, we have used openbabel, which utilizes the openly available InChI Trust software library, to produce InChI strings for all compound entries that do not generate errors in both databases.  We have added these results to the manuscript, including the improved compound harmonization (3103 vs 2336 correspondences) provided by these systematically generated InChI strings. However, the compound harmonization from all consistent identifiers only improves to 5929 vs 5681 correspondences.  Also, in comparing the systematic InChI strings to those provided by each database, only 7.3% of the KEGG InChI strings and 0.4% of the MetaCyc InChI strings were inconsistent.  This is a much lower level of discrepancy than what the ATLATIS analysis of PubChem observed.  We have added these results as follows to the manuscript:

“We also generated InChI identifiers based the molfile provided for each entry in each database using Open Babel [20], which utilizes the InChI software library provided by the InChI Trust [21].  We were able to generate 16530 InChI from KEGG and 15765 InChI from MetaCyc, providing 3103 correspondences.  When combined with ChEBI and KEGG Compound IDs, a total of 5929 consistent correspondences were detected…. Such errors are illustrated by the 964 out of 13216 KEGG compound entries with InChI that are inconsistent with the InChI generated from their associated molfile, representing 7.3% of the InChI-containing entries in KEGG.  Likewise, 55 out of 15076 MetaCyc compound entries have InChI that are inconsistent with the InChI generated from their associated molfile, representing 0.4% of the InChI-containing entries in MetaCyc.”

Issue 2:

First, the mathematical and computational challenges of molecular graph theory has been captured in a recent book by Erica Flapan (When Topology Meets Chemistry. A Topological Look at Molecular Chirality by Erica Flapan).  The rigorous language in this book goes a long way toward clarifying the issues of symmetry.  For example, the rigorous definition of symmetry is given - in simple terms:

point groups < molecular symmetry group < topological symmetry group < automorphism group (< means contained in)

Point groups: automorphisms induced by rotations and reflections

Molecular symmetry groups: automorphisms induced by molecular motions

Topological symmetry groups: automorphisms induced by homeomorphisms of space

Automorphism groups: automorphisms of the abstract graph

Response:

This book deals with topological concepts of molecular symmetry and asymmetry and goes far beyond the point groups interpretation of molecular symmetry that we are dealing with in this work.  However, we do understand the reviewer’s very good suggestion that we need to provide a clear definition of the type of molecular symmetry we are discussing.  Towards this end, we have made the following changes to the manuscript:

“However, more complex molecular symmetries like that illustrated by KEGG C04167 will require the use of algorithms that can detect all possible molecular symmetries (i.e. automorphisms induced by rotations and reflections of the Â3 embedded graph) using a 3-dimensional representation of the compound [30].”

Issue 3:

Second, the following three references should be added:

Methods and algorithms III. Graph invariants and stabilization methods

  1. Tinhofer and M. Klin, “Algebraic combinatorics in mathematical chemistry. Tech. Rep. TUM -M9902, Technische Universitat Munchen, 1999.

Get Your Atoms in Order—An Open-Source Implementation of a Novel and Robust Molecular Canonicalization Algorithm

Nadine Schneider, Roger A. Sayle, and Gregory A. Landrum

Journal of Chemical Information and Modeling 2015 55 (10), 2111-2120

DOI: 10.1021/acs.jcim.5b00543

ArbAlign: A Tool for Optimal Alignment of Arbitrarily Ordered Isomers Using the Kuhn–Munkres Algorithm

Berhane Temelso, Joel M. Mabey, Toshiro Kubota, Nana Appiah-Padi, and George C. Shields

Journal of Chemical Information and Modeling 2017 57 (5), 1045-1054

DOI: 10.1021/acs.jcim.6b00546

Response:

We understand the reviewer’s suggestion of providing a context of our methods with respect to previously published methods.  However, we did not set out to create a new molecular canonicalization algorithm, even though our algorithm has many of the qualities of a molecular canonicalization algorithm.  We were developing an algorithm that facilitates harmonization of compounds across two metabolic network databases.

With that said, we recognize the importance of putting our work within the context of prior research. Towards this end, we have added the following to the manuscript, which includes the first two references.  However, the third reference is actually not directly relevant, which is why we have left it out.

“While the molecular graph coloring method has similarities to molecular canonicalization methods [21, 31, 32], it was designed to facilitate harmonization of compounds between metabolic databases.”

Issue 4:

Finally, with respect to the analysis of metabolic pathways and issues related to graph theory, the joint work of IUBMB and IUPAC should be acknowledged

Response:

We thank the reviewer for pointing this out.  We have added the following references acknowledging the work of IUBMB and IUPAC in the development of EC numbers:

Danchin, A. "Enzyme nomenclature, recommendations (1992) of the nomenclature committee or the international union of biochemistry and molecular biology: edited by EC Webb, Harcourt Brace Jovanovich Inc, 1992." Biochimie 75.6 (1993): 501.

McDonald, Andrew G., Sinead Boyce, and Keith F. Tipton. "ExplorEnz: the primary source of the IUBMB enzyme list." Nucleic acids research 37.suppl_1 (2009): D593-D597.

“The Enzyme Commission (EC) number is a numerical classification scheme for enzymes, playing a key role in classifying enzymatic reactions [27-28].”